# Prevalence of Anti-SARS-CoV-2 Antibodies and Potential Determinants among the Belgian Adult Population: Baseline Results of a Prospective Cohort Study

**DOI:** 10.3390/v14050920

**Published:** 2022-04-28

**Authors:** Victoria Leclercq, Nayema Van den Houte, Lydia Gisle, Inge Roukaerts, Cyril Barbezange, Isabelle Desombere, Els Duysburgh, Johan Van der Heyden

**Affiliations:** 1Department of Epidemiology and Public Health, Sciensano, 1050 Brussels, Belgium; nayema.vandenhoute@sciensano.be (N.V.d.H.); lydia.gisle@sciensano.be (L.G.); elza.duysburgh@sciensano.be (E.D.); johan.vanderheyden@sciensano.be (J.V.d.H.); 2Department of Infectious Diseases in Humans, Sciensano, 1050 Brussels, Belgium; inge.roukaerts@sciensano.be (I.R.); cyril.barbezange@sciensano.be (C.B.); isabelle.desombere@sciensano.be (I.D.)

**Keywords:** SARS-CoV-2, COVID-19, antibodies, seroprevalence, cohort, population-based study, Belgium

## Abstract

The prevalence of anti-SARS-CoV-2 antibodies and potential determinants were assessed in a random sample representative of the Belgian adult population. In total, 14,201 individuals (≥18 years) were invited by mail to provide saliva via an Oracol^®^ swab. Survey weights were applied, and potential determinants were estimated using multivariable logistic regressions. Between March and August 2021, 2767 individuals participated in the first data collection. During this period, which coincided with the onset of the vaccination campaign, the seroprevalence in the population increased from 25.2% in March/April to 78.1% in July. Among the vaccinated there was an increase from 74,2% to 98.8%; among the unvaccinated, the seroprevalence remained stable (around 17%). Among the vaccinated, factors significantly associated with the presence of antibodies were: having at least one chronic disease (OR_a_ 0.22 (95% CI 0.08–0.62)), having received an mRNA-type vaccine (OR_a_ 5.38 (95% CI 1.72–16.80)), and having received an influenza vaccine in 2020–2021 (OR_a_ 3.79 (95% CI 1.30–11.07)). Among the unvaccinated, having a non-O blood type (OR_a_ 2.00 (95% CI 1.09–3.67)) and having one or more positive COVID-19 tests (OR_a_ 11.04 (95% CI 4.69–26.02)) were significantly associated. This study provides a better understanding of vaccine- and/or natural-induced presence of anti-SARS-CoV-2 antibodies and factors that are associated with this presence.

## 1. Introduction

On 11 March 2020, the World Health Organization (WHO) declared the severe acute respiratory syndrome coronavirus 2 (SARS-CoV-2) outbreak as a pandemic [1]. In Belgium, the first confirmed SARS-CoV-2 case was reported on 2 February 2020 [2]. From then, the virus spread rapidly, causing a high number of infections, hospitalizations, and casualties [3].

Since the beginning of the pandemic, large efforts have been directed towards finding a treatment and vaccine. The first vaccines against SARS-CoV-2 became available in December 2020 and in January 2021, the Belgian vaccination campaign was launched, starting with the vaccination of priority groups as defined by the Belgian Superior Health Council, being health care workers, people aged 65 years and above, and people younger than 65 years at risk of developing a severe form of COVID-19 due to existing comorbidities [4].

Epidemiological surveillance to monitor the COVID-19 epidemic were established. This included a systematic collection, analysis, and interpretation of the number of COVID-19 cases, hospitalizations, ICU patients, deaths, and a follow up of the vaccination rate [5,6]. Additionally, the knowledge on the prevalence of anti-SARS-CoV-2 antibodies in the population is important to better understand the epidemiological situation and provide, among others, valuable information regarding vaccine-induced and/or natural immunity against SARS-CoV-2 infections [7]. Furthermore, this enhances the understanding of the antibody response following vaccination by identifying factors that may explain the presence or absence of antibodies, the duration of the presence of these antibodies, or the protection against other infections. Monitoring the presence of antibodies among unvaccinated people further allows a better estimation of the virus circulation by identifying, for example, infected but asymptomatic subjects, potentially missed by the usual test and tracing system.

COVID-19 seroprevalence studies have been conducted in several countries; as, among others, in Switzerland [8], Portugal [9], Austria [10], and England [11]. In Belgium, studies have first been set up in specific population groups such as blood donors [12], hospital healthcare workers [13], primary healthcare workers [14], schoolchildren and school staff [15], and residents and staff in nursing homes [16]. The SalivaHIS study is complementary to these studies as it assesses the prevalence of anti-SARS-2-CoV antibodies in the community dwelling population aged 18 years and older. In the SalivaHIS study, the presence of anti-SARS-CoV-2 antibodies is measured in saliva.

Seroprevalence studies are mostly performed using serologic assays detecting anti-SARS-CoV-2 antibodies in serum or blood [17]. These methods are the reference to assess if a person had antibodies against COVID-19 or not. However, serological-based studies may be more difficult to implement in the general population because of logistical and practical constraints to obtain a serum sample in a geographically scattered random population. To overcome this issue, we considered the detection of antibodies in the saliva as an acceptable non-invasive alternative to serological testing to monitor SARS-CoV-2 antibody development [18,19]. Experiences in the US and Belgium showed that the use of saliva-based antibody testing is a scalable alternative to blood-based antibody testing [18,20].

This study aims to assess the prevalence of anti-SARS-CoV-2 antibodies among the general adult population in Belgium during the period March–August 2021 and to investigate to what extent this prevalence varies in function of socio-demographic characteristics, health status, and COVID-19-related health behavior, both among the vaccinated and unvaccinated populations.

## 2. Materials and Methods

We followed for this paper the Strengthening the Reporting of Observational Studies in Epidemiology (STROBE) [21] recommendations. The study research protocol is available on the Open Science Framework platform: https://osf.io/5tf8s/ (accessed on 27 April 2022).

### 2.1. Design, Setting, and Study Population

The SalivaHIS study is a population-based prospective cohort study in which a saliva sample and questionnaire data are self-collected from a sample of the Belgian adult population.

This paper reports the results of the first testing period which took place between 25 March and 31 August 2021. Additional testing periods are planned after 3 and 6 months. Findings reported in this paper do not include data from the follow-up testing periods; hence, this study has a cross-sectional design.

The target population consists of people aged 18 years and above residing in private dwellings in Belgium without any restrictions and regardless of their use of health services. The sample frame is the National Register, which includes all citizens with an official address in Belgium. The primary selection unit was the household and all residents aged 18 years and older of the selected households were eligible for participation. Households were selected through a stratified random sampling method using 24 strata defined by the region of residence (Flanders, Brussels, capital, and Wallonia), sex, and age group of the reference person of the household, as defined in the National Register.

The fieldwork had a pilot phase (from 29 March to 5 May 2021) including 1339 individuals belonging to 634 households and the main study phase (from 17 May to 28 August 2021) including 12,862 individuals belonging to 7598 households. Strata information on the participation rate by stratum in the pilot phase was used to decide on the number of households to be invited per stratum for the main study. As the study procedures did not change between the pilot phase and the main study, the participants of the pilot phase were integrated into the main study sample.

### 2.2. Data Collection

People selected for the study received an invitation letter by regular mail, two consent forms, an Oracol^®^ tube (Malvern Medical Developments Ltd., Worcester, Worcestershire, UK) to collect saliva, an instruction document on how to collect the saliva and how to obtain test results, a paper questionnaire, and a prepaid return envelope. An identification code was assigned to all participants to create pseudonymity and to link the different data collection materials. For privacy reasons, researchers had no access to any personal information of the respondents and invitations were sent through a trusted third party, Statbel, the Belgian national office of statistics. To participate in the study, it was required to send back a saliva sample, a consent form, and a completed questionnaire. It was also possible to complete the questionnaire online. The topics in the questionnaire selected based on their relevance for the analysis of determinants of COVID-19 seroprevalence status were: (1) sociodemographic information, (2) presence of chronic diseases, (3) occupational status, (4) financial situation, (5) access to health care services, (6) mental health, (7) social contacts, (8) lifestyle, (9) possible contact with SARS-CoV-2 virus and consequences, (10) adherence to corona measures, (11) vaccination status, and (12) attitude towards vaccination.

Invitations for the pilot phase were sent on 25 March 2021, invitations for the main study between 18 May and 15 June 2021 in three batches. The spread of the mailing over a period of 4 weeks was done to be able to assess the evolution of the outcome indicators across the study period.

### 2.3. Outcomes and Potential Determinants

The main outcome was the presence of anti-SARS-CoV-2 antibodies among the general population. The saliva samples were analyzed by the laboratories in Sciensano. The detection of anti-SARS-CoV-2 antibodies in saliva was completed using the WANTAI SARS-CoV-2 IgG ELISA (Wantai Bio-Pharm, cat n° WS-1396), a semi-quantitative measure of anti-RBD (Receptor Binding Domain) IgG, customized for saliva (in house protocol). The cut-off value for anti-RBD IgG positivity in saliva was established using well-characterized PCR-confirmed samples from adults of which corresponding serum and saliva were available. Using the predefined positivity cut off (cut off value > 2), a binary result was provided by the laboratory, classifying each sample as positive versus negative for the presence of antibodies against SARS-CoV-2. This cut-off resulted in a specificity of 98.9% and a sensitivity of 91.8%.

Vaccination status was assessed using questionnaire data. At the time of the study period, four vaccine types were administered in Belgium: AstraZeneca/ChAdOx1-S, Pfizer/BNT1272b2, Moderna/mRNA-1273, and Janssen/Ad26.COV2.S. People were considered as fully vaccinated if they had received, since at least two weeks, one dose of the Janssen/Ad26.COV2.S vaccine or two doses of the other vaccines. The questionnaire data also allowed to identify people who were aware of having had a close contact with a COVID-19 positive person and those who had one or more positive COVID-19 test results themselves.

Determinants that were hypothesized to potentially affect the presence of anti-SARS-CoV-2 antibodies included demographic characteristics (age, gender, region of residence), living situation, household size, education level, and occupation status. Health-related and biological factors which may also be relevant for infection with the SARS-CoV-2 virus were addressed.

Overweight was defined as a body mass index (BMI) ≥ 25 kg/m^2^ and obesity as a BMI ≥ 30 kg/m^2^. Chronic diseases that were listed included diseases to which the Belgian Superior Health Council allocated a priority status regarding vaccination, being: chronic lung disease, chronic cardiovascular disease, diabetes, chronic neurological disease, cancer (including blood cancer), chronic renal disease, immune compromised (except HIV), and transplant patients. An indicator was created identifying people with at least one chronic disease and one with at least two chronic diseases.

Health-related behavior considered in this study included smoking status and adherence to public health preventive measures implemented in the framework of the corona crisis. Nine common COVID-19 measures were taken into consideration. People who indicated having strictly complied with all nine measures were categorized as having “strictly followed them”; people who reported they strictly complied with 5–8 out of the 9 measures were categorized as having “moderately followed them”; and those who complied with less than 5 measures were referred as having “insufficiently followed the measures”.

### 2.4. Sample Size

The main outcome indicator of the study was the prevalence of anti-SARS-CoV-2 antibodies among the general population. Sample size calculations indicated that a sample size of 1200 individuals in each of the 3 Belgian regions would be large enough to obtain regional estimates on the main outcome indicator with a sufficiently high precision. More specifically, assuming a prevalence of anti-SARS-Cov-2 antibodies in the population of 50% and taking into account an alpha error of 0.05, these sample sizes would yield a margin of error of 1.9% at the level of Belgium and 3.3% at the level of the regions.

### 2.5. Time Periods Considered for Trend Analyses

Saliva samples were returned to the lab between 29 March (week 13) and 27 August 2021 (week 34). Figure 1 shows that the collection of saliva samples was not equally spread over this study period. Too few saliva samples were collected in the weeks between April 12 and May 16, corresponding to the period between the pilot phase and the launch of the main study, and after July 11 to have sufficiently precise estimates for those time periods. It was therefore decided to focus for the trend analysis on five periods of two weeks for which at least 200 saliva samples were collected (Table 1).

### 2.6. Statistical Analyses

Initial weighted seroprevalence estimates were calculated with 95% confidence intervals, taking into account the design of the survey. Post stratification survey weights were calculated based on the population structure on the 1 January 2021 obtained from Statbel as the auxiliary reference database. The effect of the household cluster was also taken into account in the survey design settings.

Correlates of the presence of anti-SARS-CoV-2 antibodies were assessed though logistic regression analysis. Because associations between having anti-SARS-CoV-2 antibodies and other potential determinants may differ for the vaccinated and the unvaccinated populations, the choice of variables to be included in the models was not the same in both groups. In the model exploring determinants of seroprevalence among the vaccinated population, variables that could influence the occurrence of antibodies in this particular population were explored (e.g., type of vaccine, presence of chronic disease, etc.). The presence of antibodies among the unvaccinated population is directly linked to the exposure to the virus. In the model exploring determinants of seroprevalence in this population, other variables were explored, such as history of a COVID-19 infection, contact with a positive person or compliance with preventive measures against the spread of the virus.

To assess time trends during the study period, data collections were analyzed as repeated cross-sectional data. Post stratification survey weights were calculated for each two-week period separately. To take into account a differential participation rate between vaccinated and unvaccinated people, weights to assess the trends in the prevalence of anti-SARS-CoV-2 antibodies took into account regional, age, and gender differences between the sample and the general population, but also the differences in the vaccination status at each of the given periods. This was completed by multiplying the initial total sample weight with a correction factor. The correction factor was obtained by dividing the number of people by region and vaccination status in the population by the number of participants by region and vaccination status in the sample. This way, the sample distribution weighted by region and vaccination status matched this distribution in the general population.

Logistic regression analysis was applied to assess time trends between the 5 time-periods. The longer time lag between the first two-week period (week 13–14) and the four other periods (week 20–27) was taken into account in the analysis.

## 3. Results

### 3.1. Description of the Population

Figure 2 gives an overview of the number of individuals invited, the participation rate and the net sample obtained in the pilot phase and in the main study for the three Belgian Regions. A total of 14,201 individuals were invited to participate in the SalivaHIS study. At the first testing period, the study included 2767 participants (with a participation rate of 19.5%) of which 2288 (82.6%) had a saliva sample with sufficient salivary volume to ensure high quality analysis.

An overview of the characteristics of the study population and their vaccination status is presented in Table 2. In summary, 54.9% of the participants were women, 46.4% were between 18 and 49 years old, 42.7% had at most a high school diploma, and 8.1% were a healthcare worker. As a result of the Belgian vaccination strategy at the time of the study, which focused on priority groups, vaccinated people were relatively older and included a higher share of women, health care workers, people with at least one chronic disease, and people vaccinated against influenza. As compared to the vaccinated, the unvaccinated were younger, more often men and smokers, as well as people who reported to have tested positive for COVID-19.

### 3.2. Prevalence of Anti-SARS-CoV-2 Antibodies over Time among the Study Population and by Vaccination Status

Figure 3 presents the prevalence of anti-SARS-CoV-2 antibodies among the total study population and by vaccination status, by two-week periods of saliva reception as a cross-sectional collection. Among the total study population, the prevalence of anti-SARS-CoV-2 antibodies increased significantly from 25.2% (95% CI: 18.8–31.6) in the first period (week 13–14) to 78.1% (95% CI: 69.2–87.0) in the last period studied (week 26–27). Among the unvaccinated, a not significant increase in the prevalence of anti-SARS-CoV-2 antibodies was observed between the fourth study period (week 24–25: 16.8% 95% CI: 9.8–23.7) and the last study period (week 26–27: 28.9% (95% CI: 4.0–53.7). However, caution is needed to interpret this increase, because the number of saliva samples for the last study period was rather low. Among the vaccinated, from the second period onwards, the prevalence of anti-SARS-CoV-2 antibodies was above 90% and stable (ranging between 93.1% (95% CI 88.3–97.8) and 97.9% (95% CI 93.9–100)). At the first period, this prevalence was remarkably lower (week 13–14: 74.2% (95% CI 52.2–96.1)).

### 3.3. Determinants of the Prevalence of Antibodies among Vaccinated and Unvaccinated Populations

Table 3 shows the prevalence of anti-SARS-CoV-2 antibodies and the results of the logistic regression investigating determinants with having antibodies for both vaccinated and unvaccinated populations.

#### 3.3.1. Vaccinated Population

With the exception of those vaccinated with an adenoviral-vectored vaccine type (i.e., AstraZeneca/ChAdOx1-S and Janssen/Ad26.COV2.S vaccine), the prevalence of anti-SARS-CoV-2 antibodies was over 90% among the vaccinated population. In the univariate analyses, no significant differences in the prevalence of anti-SARS-CoV-2 antibodies were observed by age, sex, region of residence, level of education, household size, being a health care worker or not, being overweight or obese, or the number of days since the last COVID-19 vaccination. The odds of having anti-SARS-CoV-2 antibodies was significantly lower in vaccinated people with a least one chronic disease (OR 0.30 (95% CI 0.12–0.76)) compared to those without chronic disease. The odds of having antibodies were significantly higher in people who had received an mRNA vaccine compared to those who received an adenoviral-vectored vaccine (OR 5.10 (95% CI 1.89–13.79)) and in people who had been vaccinated against influenza in 2020–2021 (OR 2.62 (95% CI 1.06–6.49)) compared to those who are not vaccinated against influenza. All variables found to be significantly associated with having anti-SARS-CoV-2 antibodies in the univariate analyses were modeled in a multivariable model. After adjustment for confounding, the three same variables remained significantly associated with having anti-SARS-CoV-2 antibodies.

#### 3.3.2. Unvaccinated Population

Not surprisingly, the prevalence of anti-SARS-CoV-2 antibodies is lower among the unvaccinated. Interestingly, only 68.7% of the unvaccinated people with one or more positive COVID-19 tests and 11.1% without a positive COVID-19 test presented antibodies. From the univariate analyses, it appears that the odds of being a woman (OR 1.46 (95% CI 1.01–2.12)), having a non-O blood type (OR 1.83 (95% CI 1.11–3.01)), having a close contact with a COVID-19 positive person (OR 3.28 (95% CI 2.08–5.17)), and having one or more positive COVID-19 tests (OR 17.62 (95% CI 10.16–30.57)) were positively associated with having anti-SARS-CoV-2 antibodies. Being a smoker (OR 0.36 (95% CI 0.18–0.72)) was negatively associated with having anti-SARS-CoV-2 antibodies. No association between the presence of anti-SARS-CoV-2 antibodies and age, region of residence, level of education, number of persons in the household, being a health care worker or not, presence of a chronic disease, being overweight or obese, having a rhesus type, having received an influenza vaccine, or following the preventive measure could be established. As for the analyses in the vaccinated population, all variables found to be significantly associated with having antibodies in the univariate analyses were modeled in a multivariable equation. After adjustment for confounding, results showed that having a non-O blood type (OR 2.00 (95% CI 1.09–3.67)) and having one or more positive COVID-19 tests (OR 11.04 (95% CI 4.69–26.02)) remained positively associated with having antibodies.

## 4. Discussion

Results from the SalivaHIS study, conducted when the vaccination campaign in Belgium was running at full speed (between March and August 2021), found that the prevalence of anti-SARS-CoV-2 antibodies among the general population in Belgium increased significantly from 25.2% in the first period (March 2021, week 13–14) to 78.1% in the last period (July 2021, week 26–27). A previous study, conducted after the first national lockdown in Belgium, showed a seroprevalence of 4.2% in October 2020 [22]. Furthermore, just before the launch of the vaccination campaign in January 2021, seroprevalence studies among Belgian blood donors [12] and primary health care workers [12] described a seroprevalence of 18.7% and 15.1%, respectively. The overall higher seroprevalence in our study mainly reflects the success of the ongoing vaccination campaign. However, also among the unvaccinated population we found a substantially increased seroprevalence compared to October 2020. This shows that there has been an important virus circulation in Belgium during the autumn of 2020, which is compatible with the second wave during this period, and the third wave (March–April 2021). It is remarkable that the seroprevalence in the unvaccinated population remained quite stable (around 17%) between 27 March (start week 13) and 27 June (end week 25). No apparent reason can be found for the sharp increase in the seroprevalence between week 24–25 and week 26–27. This increase is not significant and probably an artefact since the number of saliva samples for this last study period was rather low. Regarding seroprevalence among the vaccinated population, the lower percentage in the beginning of April could be related to the fact that at that time the majority of vaccinated people were older people, with a lower immune response than younger people. During this first period, vaccination in Belgium focused mainly on the older population, people with comorbidities, and health care workers.

SalivaHIS results were compared with other national and international seroprevalence studies that were conducted during the same study period. From these studies it appears that the seroprevalence was 66.1% in the general population in Geneva (July 2021) [8], 89.4% in the general population in England (July 2021) [11], 75.9% among blood donors in Austria (July 2021) [10], and 98.0% (July 2021) among Belgian blood donors [12]. The seroprevalence of 78.1% (July 2021) observed in our study is, thus, relatively consistent with what was found in those other studies [8,10,11,12]. Of course, these results should be interpreted with caution as the epidemiological situation and the vaccination campaign were different in each country.

Our study provides some interesting results regarding the prevalence of anti-SARS-CoV-2 antibodies among the unvaccinated population in relation to a previous COVID-19 infection. About 18% of the unvaccinated population showed anti-SARS-CoV-2 antibodies. It is important to note that the first data collection of this study took place before the fourth (from 4 October 2021 to 26 December 2021) and fifth (from 27 December 2021 to the end of February 2022) COVID-19 waves in Belgium including the emergence of the Omicron variant [23]. The seroprevalence of 18% is similar to the seroprevalence found at the same time among unvaccinated school children and staff [12] during the same period. It is striking that the seroprevalence among unvaccinated participants with one or more COVID-19 positive tests was only 68.7%. This observation is different than the one identified in a Austrian seroprevalence study among unvaccinated blood donors [10] and in a US study evaluating the prevalence of antibodies among unvaccinated US adults [24]. The lower seroprevalence identified in this study could be explained by the fact that the time since the infection was not considered in our analyses. The infection could have occurred up to 12 months before the saliva collection. Therefore, we can hypothesize a decrease in antibodies after SARS-CoV-2 infection over time. In the Austrian study, they identified a substantial decline in antibody levels during the first 6 months post-infection and a slower decline in the subsequent months [10]. However, a decrease in antibodies after infection is not always observed in other studies [24,25]. A higher seroprevalence among vaccinated people than among people who had a COVID-19 infection could suggest that vaccine-induced antibodies are more effective than natural immunity, but this needs to be interpreted with caution [26]. To investigate this further, the time since vaccination and infection should also be taken into account. Furthermore, in this study as in the Austrian study, around 11% of unvaccinated participants without a previous COVID-19 infection had developed antibodies without knowing it [10]. Unvaccinated people without a known COVID-19 infection but with anti-SARS-CoV-2 antibodies are either people who had an asymptomatic infection or people who had mild symptoms but remained undiagnosed. They are missed by the epidemiological surveillance which leads to an underestimation of the true number of infected people. This finding emphasizes the added value of seroprevalence studies which provide information on the epidemic, taking into account both diagnosed and undiagnosed cases.

While correlates of protection for SARS-CoV-2 are still not well defined, it is generally accepted that levels of antibodies targeting the Spike RBD of SARS-CoV-2 correlate with a certain level of protection [17,27]. The results of this study identified some predictive factors associated with the prevalence of anti-SARS-CoV-2 antibodies among the vaccinated and unvaccinated populations.

Although the large majority of the fully vaccinated population in our study had anti-SARS-CoV-2 antibodies, 5% remained non-responders after vaccination. Few data are available on the factors associated with the absence of anti-SARS-CoV-2 antibodies following vaccination against COVID-19. Therefore, an individual’s response after vaccination cannot be predicted yet. Our results found that the seroprevalence was significantly lower in participants with at least one chronic disease, even after adjustment with covariates. Other research is needed to explore if this could be related to a lower immune response of people with certain chronic diseases. Some preliminary results support the fact that the presence of anti-SARS-CoV-2 antibodies could be lower among a population subgroup with chronic diseases and particularly in the immunocompromised population [5,28]. Unfortunately, the exploration of more in-depth data in this study is limited by small numbers of observations among the different chronic diseases. Since the people with at least one chronic disease are considered a priority for vaccination in Belgium, the number of days since last vaccination could influence seroprevalence among these people. However, in this study, there is no difference in number of days since last vaccination between people with and without a chronic disease (OR: 1.00 (95% CI 0.99–1.02), *p*-value: 0.49). A higher seroprevalence was associated with being vaccinated against influenza during the previous vaccination season and being vaccinated with an mRNA vaccine compared to an adenoviral-vectored vaccine. Although the potential role of influenza vaccination on COVID-19 outcomes is still unclear, some other studies underlined a negative association between SARS-CoV-2 infection and having been previously vaccinated against the flu [29,30]. However, further research is needed to understand the protective role of a previous influenza vaccination. With respect to the disparity in antibody presence between the two types of vaccines, other studies have also shown a more consistent and higher presence of anti-SARS-CoV-2 antibodies in people who received an mRNA vaccine type [5,31].

Initial evidence, although often contradictory, shows that unvaccinated people with blood type A are at higher risk of being infected with SARS-CoV-2, while the risk is decreased in people with blood type O [31,32,33]. In line with the first evidence, results of this study show that people with blood types A, B, or AB are positively associated with the presence of antibodies. However, while some studies have shown an association between SARS-CoV-2 infection and the rhesus-positive blood type [32,33,34], these results are not confirmed in the present study. The strong association between a prior infection with SARS-CoV-2 and a higher presence of anti-SARS-CoV-2 antibodies is in line with what can be expected and consistent with the literature [10,24]. In contrast to findings from other studies, the results of our study, after adjustment for cofounding factors, did not highlight a significant association between the presence of anti-SARS-CoV-2 and different risk factors such as age [8,10,35], sex [35,36], overweight [10,35], or smoking status [10].

Some limitations to this study need to be highlighted. First, the presence of antibodies was determined by a salivary test and not by a regular serum test. Here, a salivary test was chosen because it is a non-invasive method for comprehensive determination of the presence of vaccine- or natural-induced SARS-CoV-2 antibodies, and facilitates large-scale serosurveillance to assess population seropositivity. The use of a salivary test, although less sensitive than a serum test, is generally accepted for serosurveillance studies and has been used and validated in other countries [18] and for other pathogens. Unfortunately, the multiplex ELISA test which would allow to make such a distinction was not validated for use in saliva; hence, it was not possible to determine in this study whether vaccinated people with a positive test result had been infected or not. Furthermore, the fact that the higher prevalence among people previously vaccinated against influenza and among people vaccinated with an mRNA vaccine, observed in studies based on blood samples, is also captured by our study confirms the validity of a saliva-based seroprevalence study. Second, saliva was self-collected by the participants with the Oracol device. Despite an information leaflet and instruction video on how to collect the sample, 17% of the samples contained too little saliva to be analyzed. We have, however, no reason to believe that the participants for whom no saliva result was available, systematically differed from those for whom a result was available. Third, although performance characteristics of the WANTAI SARS-CoV-2 RBD IgG ELISA were good (in house protocol, sensitivity of 91.8% and a specificity of 98.91% in adults), an information bias due to false negative or false positive test results cannot be excluded. Furthermore, the saliva SARS-CoV-2 RBD IgG ELISA returns a semi-quantitative result according to a predefined threshold but did not allow to define a quantitative level of antibodies. Fourth, the required sample size of 1200 individuals per region estimated in the research protocol was not reached because the participation rate was lower than anticipated. However, the net obtained sample was still acceptable for seroprevalence estimates with reasonable precision. A major strength of this study lies in the fact that the SalivaHIS sample was a population-based sample and post-stratification weights were calculated with the aim to produce estimates which were as representative as possible of the total Belgian population. However, participation in the SalivaHIS study was voluntary and although the application of weights ensured that the weighted distributions of the sample by age group, gender, and region matched exactly these distributions in the Belgian population, this could not exclude a selection bias resulting from a differential participation rate within different population groups.

The results obtained in the present and in other studies [37,38] confirm the link between the presence of antibodies and vaccination. However, knowledge on the underlying mechanism explaining the protective role of anti-SARS-CoV-2 antibodies, vaccine- or infection-induced, on infection, reinfection, or disease status remains scarce [39,40]. The emergence of SARS-CoV-2 variants of concern demonstrates that variation in transmissibility, severity, and immune escape potentially underscore the need for additional population level studies evaluating the protective role of variant-specific antibodies [38,39]. Moreover, the finding that anti-SARS-CoV-2 antibodies rapidly wane over time [40,41,42,43] drastically impacts the risk of (re)-infection and transmission. Therefore, 3 and 6 month follow-up data from our SalivaHIS cohort will allow an in-depth exploration of the determinants of seroconversion as well as the role of anti-SARS-CoV-2 antibodies on infections and their severity.

## 5. Conclusions

While the vaccination campaign was going full speed in Belgium, a large increase in the presence of anti-SARS-CoV-2 antibodies has been observed in the general population. The prevalence of antibodies was above 90% among the vaccinated population. Nearly one fifth of the unvaccinated people presented antibodies and this seroprevalence was relatively stable over the time period. Only 68.7% of previously infected, unvaccinated subjects presented antibodies. This study provides a better understanding of the acquired immunity against SARS-CoV-2 among the Belgian population. However, the presence of antibodies and potential correlates of protection against future SARS-CoV-2 infections is still unclear, specifically with emerging variants. A better understanding of the role of antibodies in the protection from the disease is necessary.

## Figures and Tables

**Figure 1 viruses-14-00920-f001:**
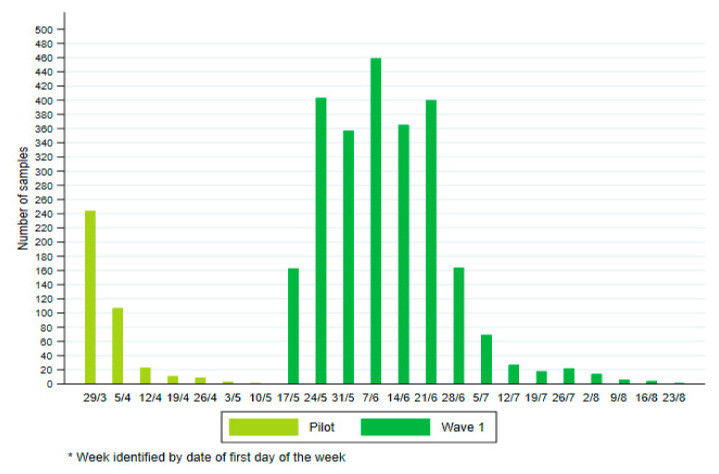
Number of saliva samples collected in the SalivaHIS study per week.

**Figure 2 viruses-14-00920-f002:**
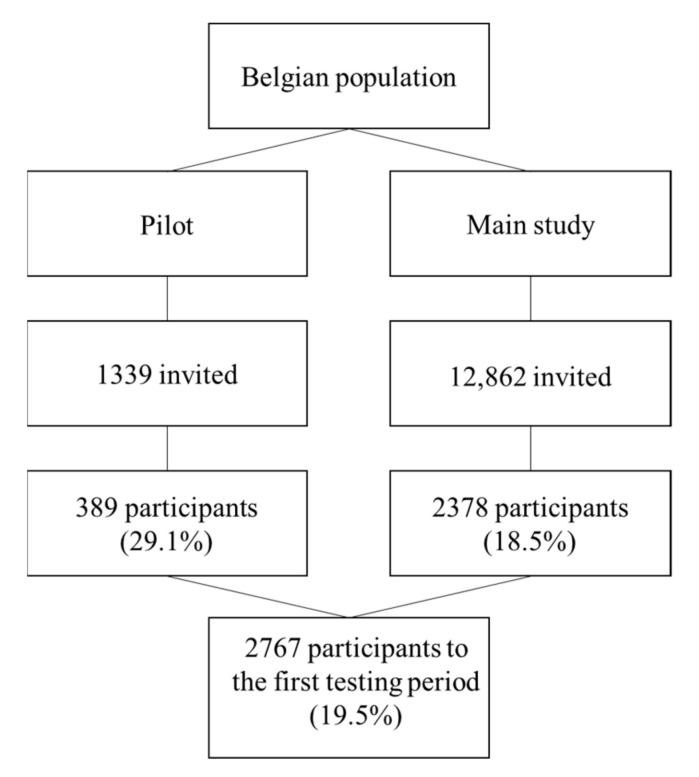
Number of invited people, participation rate, and net sample obtained.

**Figure 3 viruses-14-00920-f003:**
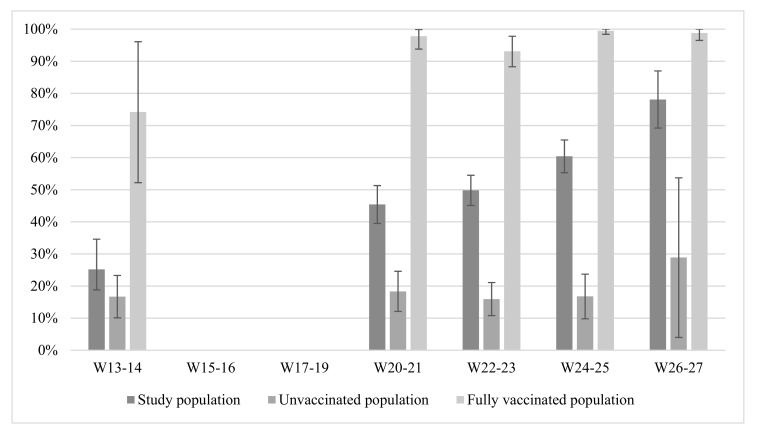
Prevalence of anti-SARS-CoV-2 antibodies by 2-week period of saliva collection.

**Table 1 viruses-14-00920-t001:** Time periods considered for trend analyses.

Periods	Week of References	Dates
Period 1	Week 13–14	29 March to 11 April 2021
Period 2	Week 20–21	17 May to 30 May 2021
Period 3	Week 22–23	31 May to 21 June 2021
Period 4	Week 24–25	14 June to 27 June 2021
Period 5	Week 26–27	28 June to 11 July 2021

**Table 2 viruses-14-00920-t002:** Description of the study participants.

		Overall Distribution		Distribution by Vaccination Status					
		Study Population N = 2767		Fully Vaccinated Population Since 2 or More Weeks N = 747 (28.5%)		Partially Vaccinated Population N = 874 (33.4%)		Unvaccinated Population N = 998 (38.1%)	
		%	Total N	%	Total N	%	Total N	%	Total N
Gender	Man	45.1	1247	39.5	295	46.5	406	46.5	464
	Woman	54.9	1520	60.5	452	53.5	468	53.5	534
Age	18–49 yrs	46.4	1285	21.6	161	35.6	311	75.1	749
	≥50 yrs	53.6	1482	78.4	586	64.4	563	24.9	249
Region	Flanders	41.9	1160	34.1	255	42.8	374	47.6	475
	Brussels	29.6	819	36.7	274	27.0	236	26.1	260
	Wallonia	28.5	788	29.2	218	30.2	264	26.4	263
Education	≤Secondary	42.7	1138	40.2	286	42.5	359	44.7	437
	Bachelor	27.1	722	29.4	209	28.8	243	24.5	239
	≥Master	30.2	807	30.5	217	28.8	243	30.8	301
Household size	1 member	15.2	382	21.6	153	15.4	129	10.4	100
	2 members	44.8	1124	54.0	383	49.4	414	34.1	327
	3 members	15.5	389	9.9	70	14.8	124	20.3	195
	≥4 members	24.4	612	14.5	103	20.4	171	35.2	338
Health care worker	Yes	8.1	214	17.8	128	5.2	43	3.5	34
	No	91.9	2423	82.2	592	94.8	786	96.5	924
Presence of at least one	Yes	24.3	643	32.8	235	25.0	207	17.1	165
chronic disease	No	75.7	2002	67.2	482	75.0	622	82.9	801
Presence of at least two	Yes	5.7	151	8.1	58	6.2	51	3.8	37
chronic diseases	No	94.3	2494	91.9	659	93.8	778	96.2	929
Overweight	Yes	48.8	1299	54.4	392	51.3	430	41.8	407
	No	51.2	1365	45.6	329	48.7	409	58.2	566
Obesity	Yes	16.6	441	21.1	152	16.7	140	12.9	126
	No	83.4	2223	78.9	569	83.3	699	87.1	847
Blood type	O blood type	56.7	1020	58.0	286	58.1	349	52.8	331
	Non-O blood type	43.3	780	42.0	207	41.9	252	47.2	296
Rhesus	Positive	81.8	1411	81.4	380	81.8	472	82.9	503
	Negative	18.2	313	18.6	87	18.2	105	17.1	104
Smokers	Yes	14.7	394	10.1	73	13.4	114	19.5	191
	No	85.3	2280	89.9	647	86.6	735	80.5	786
Influenza vaccine	Vaccinated	37.4	1011	62.7	458	42.2	360	14.4	142
	Not vaccinated	62.6	1693	37.3	273	57.8	493	85.6	844
Close contact with COVID-19 positive person	Yes	27.2	737	23.3	170	22.7	194	34.1	337
	No	56.3	1523	61.5	449	61.7	528	47.6	470
	I do not know	16.5	447	15.2	111	15.7	134	18.2	180
One or more positive COVID-19 test results	Yes	11.8	306	11.0	78	10.7	87	12.8	121
	No	88.2	2289	89.0	633	89.3	723	87.2	822
Preventive measures	Strictly followed the measures	30.3	555	38.9	189	33.6	192	21.3	148
	Moderately followed the measures	45.4	831	47.1	229	46.0	263	44.2	307
	Insufficiently followed the measures	24.3	444	14.0	68	20.5	117	34.4	239

**Table 3 viruses-14-00920-t003:** Prevalence of anti-SARS-CoV-2 antibodies among vaccinated and unvaccinated populations and potential factors associated.

		Vaccinated Population (N = 593)				Unvaccinated Population (N = 838)			
Determinants	Categories	%	Total N	Unadjusted ° OR (95% CI)	Adjusted ^$^ OR (95% CI)	%	Total N	Unadjusted ° OR (95% CI)	Adjusted ^$^ OR (95% CI)
Gender	Man	94.2	250	Ref	Ref	15.3	398	Ref	Ref
	Woman	95.6	343	1.32 (0.57–3.12)	1.09 (0.45–2.68)	21.0	440	**1.46 (1.01–2.12)** ^※^	1.33 (0.72–2.49)
Age	18–49 yrs	97.4	137	Ref	Ref	18.5	638	Ref	Ref
	≥50 yrs	94.1	456	0.43 (0.13–1.39)	0.34 (0.08–1.45)	16.1	200	0.96 (0.72–1.28)	0.87 (0.45–1.67)
Region	Flanders	97.7	196	Ref	Ref	16.4	388	Ref	Ref
	Brussels	97.8	228	2.27 (0.78–6.60)	2.07 (0.59–7.30)	22.9	225	1.52 (0.97–2.37)	0.76 (0.34–1.69)
	Wallonia	93.9	169	0.81 (0.30–2.15)	1.25 (0.40–3.90)	19.6	225	1.25 (0.74–2.10)	0.84 (0.34–2.05)
Education	≤Secondary	93.8	219	Ref		17.2	358	Ref	
	Bachelor	95.9	169	1.54 (0.51–4.70)		20.9	206	1.27 (0.80–2.02)	
	≥Master	96.4	177	1.78 (0.50–6.32)		16.8	254	0.97 (0.61–1.56)	
Household size	1 member	95.6	117	Ref		20.0	82	Ref	
	2 members	95.1	303	0.89 (0.22–3.54)		12.5	269	0.57 (0.27–1.22)	
	3 members	88.2	61	0.34 (0.07–1.65)		13.5	174	0.62 (0.27–1.44)	
	≥4 members	97.9	84	2.13 (0.29–15.84)		25.0	282	1.33 (0.64–2.77)	
Health care worker	Yes	95.2	103	0.99 (0.31–3.13)		17.4	33	0.99 (0.38–2.57)	
	No	95.2	470	Ref		17.6	770	Ref	
Presence of at least one chronic disease	Yes	90.4	173	**0.30 (0.12–0.76) ^※^**	**0.22 (0.08–0.62) ^※^**	21.0	125	1.29 (0.78–2.14)	
	No	97.0	400	Ref	Ref	17.1	684	Ref	
Presence of at least two chronic diseases	Yes	93.9	41	0.81 (0.18–3.73)		19.7	27	1.15 (0.42–3.11)	
	No	95.0	532	Ref	Ref	17.6	782	Ref	
BMI 25–30 kg/m^2^	Yes	93.8	306	0.59 (0.23–1.48)		18.5	335	1.12 (0.75–1.66)	
	No	96.3	271	Ref		16.8	481	Ref	
BMI > 30 kg/m^2^	Yes	95.0	108	1.02 (0.36–2.88)		17.2	103	0.97 (0.55–1.72)	
	No	94.9	469	Ref		17.6	713	Ref	
Blood type	O blood type					13.7	291	Ref	Ref
	Non-O blood type					22.5	236	**1.83 (1.11–3.01)** ^※^	**2.00 (1.09–3.67) ^※^**
Rhesus	Positive					15.6	422	0.56 (0.32–1.00)	
	Negative					24.9	90	Ref	
Smokers	Yes					8.5	161	**0.36 (0.18–0.72)** ^※^	0.44 (0.14–1.33)
	No					20.2	658	Ref	Ref
Close contact with COVID-19 positive person	Yes					29.9	282	**3.28 (2.08–5.17) ***	1.44 (0.71–2.92)
	No					11.5	388	Ref	Ref
	I do not know					14.2	159	1.27 (0.69–2.31)	0.64 (0.27–1.52)
One or more positive COVID-19 test results	Yes					68.7	103	**17.62 (10.16–30.57) ***	**11.04 (4.69–26.02) ***
	No					11.1	690	Ref	Ref
Preventive measures	Strictly followed the measures					16.1	118	Ref	
	Moderately followed the measures					19.4	261	1.26 (0.69–2.31)	
	Insufficiently followed the measures					22.3	212	1.50 (0.80–2.81)	
Influenza vaccine	Vaccinated	96.8	361	**2.62 (1.06–6.49)** ^※^	**3.79 (1.30–11.07)** ^※^	15.5	104	0.85 (0.46–1.57)	
	Not vaccinated	92.0	222	Ref	Ref	17.8	722	Ref	
Type of vaccination	mRNA vaccine	96.5	461	**5.10 (1.89–13.79) ***	**5.38 (1.72–16.80)** ^※^				
	adenoviral-vectored vaccine	84.3	58	Ref	Ref				
Number of days since the last vaccination			567	1.00 (0.99–1.02)					

° The association of each independent variable with having SARS-CoV-2 antibodies was assessed individually by univariate logistic regression. ^$^ All variables found to be significantly associated with having anti-SARS-CoV-2 antibodies in the univariate analyses were modeled in a multivariable logistic regression. ^※^ *p*-value < 0.05; * *p*-value < 0.001.

## Data Availability

Data are available on reasonable request. The statistical codes that support the findings of this study are available from the corresponding author on reasonable request.

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
