# Peer review of "Prevalence of Anti-SARS-CoV-2 Antibodies and Potential Determinants among the Belgian Adult Population: Baseline Results of a Prospective Cohort Study"

_viruses, 2022, doi:10.3390/v14050920_

Round 1
Reviewer 1 Report
In this study, Leclercq et al. report the prevalence of anti-SARS-CoV-2 antibodies in a Belgian adults population as well as determinants of the persistance of these latter antibodies. The study design is interesting regarding the sampling modality (saliva) and the results obtained are robust regarding the size of the cohort. This study was performed before the Omicron wave and, therefore, a short discussion should take place in relation to the evolution of the pandemic and which impact on this type of seroprevalence study is expected. Finally, some sentences should be rephrased (some sentences lack punctuations, like commas).
- Authors should give more details concerning the ELISA test used in the present study. According to the discussion, the used ELISA is a in-house method (this should be already described in the method section). How did you validate your "saliva cut-off" ? Moreover, the authors precise that the sensitivity was good (84.9%...) but didn't mention what was the reference method to calculate the latter sensitivity ? A commercial SARS-CoV-2 IgG assay ? Samples issued from patients with a positive RT-qPCR ?
- - L209-210: "... of which..."
- Some parts of the "Results" section should be moved to the "Material and Methods" section. For example: " Because associations between having anti-SARS-CoV-2 antibodies and other potential determinants may differ for the vaccinated and the unvaccinated population, some variables were studied only in one of the two groups. Since the presence of antibodies in vaccinated population is mainly due to vaccination, variables that could influence the occurrence of antibodies were explored (e.g.: type of vaccine, presence of chronic disease…)."
- The poor prevalence of antibodies observed in the unvaccinated people could be linked to the limitations of the collection and analytical methods used in the present study and is therefore only partially adressed as a limitation in the discussion. Moreover, the prevalence of seropositive people among the group of previously infected but unvaccinated people seems very low. Indeed, this can be highly related to the defined cut-offs of positivity, which can vary from one manufacturer to another, and thus lead to different durations of immunity before falling back below the positive threshold. My concern is that the positive cut-off defined by the authors is too high (in order to improve the specificity of the in-house ELISA).
Reviewer 2 Report
The mansucrit is very clear and very well presented.
Several limitations are listed by the authors in the discussion but my main concerns are the following
- One main limitation is that the authors did not take into account time since vaccination and it is well known that the antibody levels decrease over time. This may explained why the odds of having anti-SARS-CoV-2 antibodies was significantly lower in vaccinated people with at least one chronic disease (these people may have been vaccinated in priority, several months ago).
- It would be nice to explore the results among different chronic diseases but it seems that the numbers of observations are too small.
- I also wonder if the statistical model should include the household component.
- Some serological tests allowed the assessment of antibodies against the spike protein (anti-S) and the nucleocapsid protein (anti-N) of SARS-CoV-2 permits, in principle, to identify individuals with past infection even if they were vaccinated against COVID-19 (at least for mRNA-based vaccines).
